# An evolutionary game analysis of algorithmic indirect copyright infringement from the perspective of collusion between UGC platforms and direct infringers

**Jiangang Liu, Yuxuan Shen**◉*, **Lanlan Zhou**

.Changzhou University Business School, Jiangsu, Changzhou, China

* s22081202017@smail.cczu.edu.cn

**Data Availability Statement:** All relevant data are within the manuscript and its Supporting Information files.

## Abstract

User-generated content (UGC) is developing rapidly as an emerging platform form, however, the problem of indirect copyright infringement by algorithms is becoming more and more prominent, and infringement governance has become a key act in the development of UGC platforms. When infringement occurs, recommendation algorithms expand the scope and results of infringement, while platforms choose to conspire with direct infringers for their own interests, making it difficult for infringed persons to defend their rights. In order to analyse the influence of different factors in the platform ecosystem on the subject's behavioural strategies, a "platform-infringer" evolutionary game model is constructed, and numerical simulation is used to verify the correctness of the stable results. Based on the simulation results, it is concluded that the factors of uncertain revenue, punishment and reputation loss have important influence on the decision-making behaviour of the subject of infringement governance, and accordingly, the proposed measures on the publishers, platforms and the legal level of the government are conducive to the evolution of the system to the point of positive regulation and stability of rights protection, with a view to promoting the healthier and more stable development of the UGC platforms.

## 1 Introduction

UGC (User Generated Content) platform refers to various forms of media and creative works created by users through the Internet and its technology [1]. UGC platforms have gradually developed and flourished along with the advancement of Internet technology, especially in China, relying on demographic advantages, ushering in explosive growth, such as the explosive Jitterbug, Xiaohongshu, Himalaya, etc. Among them, short video platforms have seen the most significant growth, according to the Report. Among them, the short video platform has grown most significantly, according to the report① shows that as of June 2022, the user scale of short videos in China reached 962 million, accounting for 91.5% of Internet users as a whole. However, along with the rapid development, the problem of UGC content has gradually come to the forefront, with content that infringes on others' copyrights being particularly

**Funding:** Projects financed by the National Social Science Foundation "Research on the Governance Mechanism of Online Platform Enterprises under the Perspective of Quasi-Public Goods"(20BGLO96), awarded to JL, and by Social Science Foundation of Jiangsu Province" Game Analysis of the Governance Evolution of Webcasting Platforms from a Quasi-Public Goods Perspective."(KYCX22_2982), awarded to LZ. The funders had no role in study design, data collection and analysis, decision to publish, or preparation of the manuscript.

**Competing interests:** The authors have declared that no competing interests exist.

prevalent. For example, Aqiyi v. A well-known short video platform APP provided users with online playing and downloading services of the infringing video through various recommendation behaviours.[②] Meanwhile, in the case of Xuanting Entertainment Co. v. Himalaya Co., the court found that some of the infringing accounts' real-name information was obviously fictitious, and considered that the accounts were intentionally set up for the purpose of disseminating the novels in question, and thus the court ruled that Himalaya Co. needed to bear the responsibility for the infringement of copyright.[③] The court ruled that Himalaya was liable for the infringement. The frequent occurrence of such infringement incidents not only has a direct bearing on the survival and development of the UGC platform industry, but also makes the public indifferent to the awareness of copyright protection.

Wu Handong [2] It is proposed that in the network environment, the wide dissemination of infringing content is often the result of the joint action of the infringer and the UGC platform. That is, the infringer releases infringing content that can be disseminated, while the UGC platform provides algorithmic technical support for the dissemination of infringing content, and the combination of the two behaviours ultimately leads to the widespread dissemination of infringing content, so how to effectively govern the algorithm indirectly infringes copyright has become the primary issue for UGC platforms. When the infringed person sues the court for infringement, the court usually requires the real name information of the direct infringer, and even if the infringed person asks the platform to disclose who has committed the infringing act, the platform will not easily disclose it in most cases, so the infringed person can only sue the platform to the court. On the one hand, in order to retain more traffic through such infringing content and earn more benefits for themselves, on the other hand, in order to avoid damage to the reputation of the platform as a result of the establishment of the fact of infringement, the platform is unwilling to disclose the direct infringer's information. Infringed person usually face two major problems when defending their rights: first, it is difficult to obtain evidence, the acquisition of evidence of infringement requires the technical cooperation of the platform, and the real name information of the direct infringer needs to be provided by the platform; second, the cost is high and the benefit is low, the time and economic cost required for defending their rights is a heavy burden for the average user, and the amount of damages awarded for a successful lawsuit is generally low. Now some scholars have constructed a two-round dynamic game model through the game tree, and obtained the conclusion that litigants can improve the settlement rate [3]. Cui Li constructed a game model between platform companies, consumers and government departments and analyzed the game relationship among the three [4].Based on the perspective of collusion between UGC platforms and direct infringers, this paper analyses the game between platforms and direct infringers in the act of indirect copyright infringement by algorithms, and analyses the reasons why infringed person in the network environment do not choose to defend their rights even though they know that they are infringed.

Current research on content governance for UGC platforms is divided into two main categories, one of which is quality content incentives. Continuously acquiring high-quality UGC contributions is critical to the survival and success of the UGC industry, especially platform companies [5] A UGC platform generates multiple ways to evaluate the quality of content, the most common of which are the number of views and downloads, the number of followers a user has, and positive ratings and reviews of the content [6, 7]These approaches make it necessary for platforms to attract empowered users in terms of both content generation and consumption, and to encourage and incentivize them to participate in content production or to consume content on an ongoing basis.More relevant to this study is another category, content censorship research. Much of the discussion concerns freedom of expression, censorship, and the pros or cons of content auditing [8–10] Madio examines content audits as a tool to attract

content-sensitive advertisers and as a way to manage their ad prices [11]. Liu et al. investigated the economic incentives of social media platforms to review user-generated content by developing a theoretical model [12].

Scholar Xiao Hongjun thinks that algorithms play the role of "agent", as the "agent" of human beings, possessing part of the decision-making power that human beings "cede", so algorithms have an indirect subjectivity and can and should be socially responsible [13]. Blockchain and machine learning are important parts of the algorithms that make up the UGC platform. A systematic literature review has been conducted to analyze blockchain-based applications in various domains such as business process development [14], healthcare [15], IoT [16] and climate agriculture system [17]. Meanwhile, machine learning techniques allow us to identify the impact of online reviews and predict future customer's behaviour [18]. It has been proven that both, positive and negative comments published on comparative websites, can change the customer's opinion, which means they have great influence on the decision processes [19]. A deep learning approach to facial emotion recognition using neural network techniques has been investigated by scholars [20]. That's why algorithms should be socially responsible, but also UGC platforms have a responsibility as users of algorithms [21]. Therefore, in the online copyright infringement, the network service provider bears mainly indirect responsibility [22]. Therefore, in the online copyright infringement, the network service provider mainly bears indirect liability. Indirect infringement" in the sense of copyright law means that although the perpetrator has not carried out the act of controlling the "exclusive right" of copyright, he intentionally induces or abets others to infringe the copyright, or provides substantial help when he knows that others intend to carry out or are carrying out the infringing act [23]. Even if UGC platforms do not intentionally promote infringing content, recommendation algorithms that compete for traffic as the underlying logic will automatically give such content recommended status, which undoubtedly provides assistance to infringing behaviour and amplifies the results of infringing damages [24].

In summary the research system of algorithmic copyright infringement in the existing research has been initially formed, and gradually carried out research in many subdivided areas, which provides this paper with a solid theoretical foundation and rich research reference. However, the existing research still has the following deficiencies: (1) the research on platform governance in the existing literature mainly focuses on content governance, governance paradigm, and infringement governance, but most of the research on infringement governance is qualitative research through qualitative observation, case analysis and other methods, and fewer literature builds a mathematical model around algorithms indirectly infringing on copyright, and carries out quantitative analysis and argumentation; (2) in the existing literature on indirect copyright infringement, most of the research is on UGC platforms, which is the same as the UGC platforms. In the existing literature on indirect infringement, most of them take the UGC platform, infringer and infringed as independent subjects, and there are few studies based on the perspective of collusion between the platform and infringer. (3) Some researchers have paid attention to the game problem in the governance of subsidy and punishment of UGC platforms, but there is still very little literature on the game between platforms and infringed persons in the indirect copyright infringement of algorithms. Therefore, this paper constructs a "platform-infringed" evolutionary game model, sets up the mechanism of "collusion between the platform and the infringers", studies the game between the platform and the infringed person in the indirect copyright infringement of algorithms, analyses the motive behind the subject's behaviour, analyses the impact of various factors on analyse the motivation behind the subject's behaviour, analyse the influence of each factor on the strategic behaviour, and put forward the corresponding suggestions to control the infringement accordingly.

The structure of this paper is as follows: Section 1 introduces the research background, significance and literature review. Section 2 constructs a game model of "UGC platform and infringer". Section 3 analyzes the conditions for game equilibrium. Section 4 shows the technical implementation of the method and the analysis results. Section 5 gives conclusions and recommendations about the model and a vision for the future.

## 2 Model construction

The basic idea of evolutionary game theory originates from the dynamic evolutionary process of the biological world, which is a combination of evolutionary theory and game theory. Its research focus and method are very different from the traditional static game, and evolutionary game theory pays more attention to the dynamic process of strategy and evolutionary equilibrium rather than the result of the game [25]. Therefore, it has been widely used in multiple fields such as enterprise strategy selection, supply chain management, and system structure evolution [26–34]. In this paper, we try to use evolutionary game theory to construct a game model between UGC platform and the infringer, and explore the interactive effects of the strategic behavior of both subjects.

### 2.1 UGC platform

In the era of "traffic is king", users and even platforms themselves intentionally use technical measures and false registration to publish pirated and infringing content in order to attract traffic. At this point, platforms can choose between active regulation and collusion. First, the platform chooses the active supervision strategy, which means that the platform needs to increase the audit of user-generated content and reduce the strength of the recommendation algorithm for potentially infringing content, issue a notice and warning to the account, and restrict the traffic allocated to the account. Instead of reviewing allegedly infringing content and links after the fact [35]; second, the platform chooses the collusion strategy, which means that the platform no longer supervises the user-generated content, and only relies on the algorithmic automated decision-making to allocate the traffic, and pushes the content of the hotness, regardless of whether it is infringing. Platforms that choose a collusion strategy will be at risk of algorithmic infringement.

Assume that the probability that the platform chooses to actively regulate in the initial state is x ($0 < x < 1$), and the probability that it colludes with the infringer is $1-x$. When the platform chooses to actively regulate, the platform needs to pay a certain amount of manpower and material resources to regulate, which includes the extra cost of auditing and the unfair impact brought about by other platforms choosing to collude, so assume that the platform actively regulates with the cost of C1 [36]. And when actively regulating, there is a probability of regulatory failure of a, i.e., failure to regulate and restrict infringing content in a timely manner, leading to an infringement event. When the platform chooses the collusive strategy, the platform no longer regulates user-generated content, which not only does not incur additional costs, but also brings additional benefits to the platform E1, i.e., infringing videos bring unfair benefits to the platform.

### 2.2 Infringed persons

When infringement occurs, the infringed person can choose to defend the right and let go of two strategies. One is to defend the right, through legal channels to sue the platform or direct infringer, at this time you need to pay a certain cost of the right, including litigation costs, evidence costs, time costs, etc., if the right to defend the success of the platform will pay a certain amount of compensation for infringement. The second is to let go, at this time the infringed's

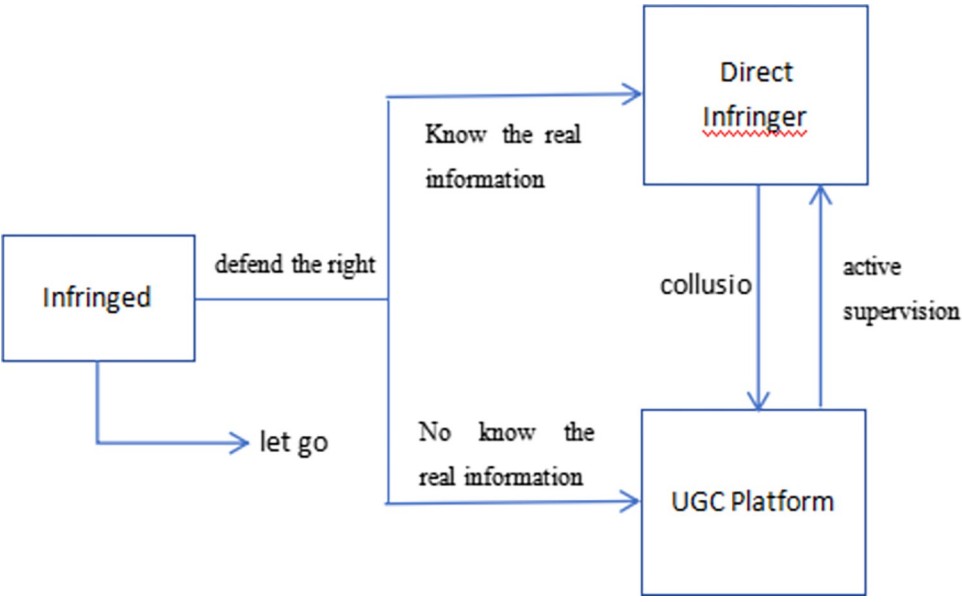

**Fig 1. Schematic diagram of subject relationship.**

work will be because of the Internet's ability to disseminate, to the infringed to bring uncertain gains, may be positive, may also be negative [37]. Positive gain means that the infringing content is viewed by more people thus bringing greater traffic attention to the infringed. Negative gains are when the pirated or secondary work takes up a larger market share, resulting in a loss of traffic to one's own work.

Assuming that the probability of the infringed choosing a rights defence strategy is y (0 < y < 1) and the probability of a laissez-faire strategy is 1—y. When the infringed chooses to defend his rights, the cost of defending his rights is C2, including the cost of time, money, and energy, etc., and the probability of success is s [38]. The infringement compensation he receives for a successful defence is F. When the infringed adopts the strategy of defending his rights and succeeds in defending his rights, the platform needs to pay the infringed the infringement compensation F, and the reputation loss caused by the infringement G. When choosing to defend the rights, the cost is certainly 0. When the infringed adopts an infringement strategy and succeeds, the platform needs to pay the infringed the infringement compensation F, as well as the reputation loss caused by the infringement G. The cost of letting go of the infringement is 0. When choosing to let go of the infringement, the infringed will be given an uncertain benefit P.

In summary, the subjects of algorithmic infringement governance include UGC platforms, infringed persons, and actual infringers of potential subjects. The subject relationship of algorithmic infringement governance is schematically shown in Fig 1.

### 2.3 Model assumptions and benefit matrix

Based on the above evolutionary game model assumptions, the relevant profit and loss variables are selected and set as shown in Table 1:

According to the relationship analysis of the game subjects and the basic assumptions of the two parties, the construction of the payoff matrix of the game subjects is shown in Table 2.

Based on the above return matrix and the probability assumptions of adopting different strategies, we can obtain the expected return $V_{11}$ when the platform chooses the active

**Table 1. Parameters and their meanings.**

| notation | hidden meaning |
|---|---|
| C1 | Extra costs when UGC platforms are actively regulated |
| E1 | UGC platforms generate additional revenue for the platform when they choose a collusion strategy |
| a | Probability of regulatory failure when actively regulating |
| C2 | Costs of defending rights when the infringed person chooses to do so |
| F | Infringement damages for the infringed by platforms that have successfully defended their rights |
| G | Reputational damage to UGC platforms after successful rights defence |
| s | Probability of success of the infringed in defending his or her rights |
| P | Uncertainty of benefit to the infringed from the event of infringement |

regulation strategy and the expected return $V_{12}$ when the collusion strategy is chosen, and the average return $V_1$ of the platform, which is:

$$V_{11} = y[-C1 + a(E1 - sF - sG)] + (1 - y)(-C1) \qquad V_{12} = y(E1 - sF - sG) + (1 - y)(E1)$$

$$V_1 = xV_{11} + (1 - x)V_{12} = (x - 1)(Fsy - E1 + Gsy) - x(C1 - aE1y + aFsy + aGsy)$$

For the infringed, the expected gain $V_{21}$ from choosing the rights defence strategy and the expected gain $V_{22}$ from choosing the laissez-faire strategy, and the average infringed's gain $V_2$, i.e:

$$V_{21} = xa(sF - C2) + (1 - x)(sF - C2) \qquad V_{22} = xP + (1 - x)P$$

$$V_2 = yV_{21} + (1 - y)V_{22} = y[(x - 1)(C2 - Fs) - ax(C2 - Fs)] - [Px - P(x - 1)](y - 1)$$

From the above conclusion, we can derive the copying dynamic equation F(x) of the platform and the copying dynamic equation F(y) of the infringed:

$$F(x) = \frac{dx}{dt} = x(V_{11} - V_1) = x(x - 1)(C1 + E1 - aE1y - Fsy - Gsy + aFsy + aGsy)$$

$$F(y) = \frac{dy}{dt} = y(V_{21} - V_2) = y(y - 1)(C2 + P - Fs - C2x + aC2x + Fsx - aFsx)$$

According to the method proposed by Friedman D [39], the local stability of the equilibrium point of the evolutionary game is judged by constructing the Jaconbian matrix of the system, and the partial derivatives of the replication dynamic equations F(x) and F(y) with respect to x and y can be obtained sequentially, which can result in the Jaconbian matrix of the evolutionary game between the ground UGC platform and the infringed person as shown in Table 3:

**Table 2. Revenue payment matrix.**

| | UGC platform | tortfeasor |
|---|---|---|
| (Positive, Rights) | -C1+a(E1-sF-sG) | a(sF-C2) |
| (active, permissive) | -C1 | aP |
| (collusion, rights) | E1-sF-sG | sF-C2 |
| (collusion, indulgence) | E1 | P |

**Table 3. Evolutionary game Jacobi matrix for platforms and the infringed.**

| x(C1+E1-aE1y-Fsy-Gsy+aFsy+aGsy)+(x-1)(C1+E1-aE1y-Fsy-Gsy+aFsy+aGsy) | -x(x-1)(aE1+Fs+Gs-aFs-aGs) |
|---|---|
| -y(y-1)(C2-aC2-Fs+aFs) | (y-1)(C2+P-Fs-C2x+aC2x+Fsx-aFsx)+y(C2+P-Fs-C2x+aC2x+Fsx-aFsx) |

## 3 Evolutionary game equilibrium analysis

In the evolutionary game between the platform and the infringed system, due to the asymmetry and incompleteness of information, each subject is limited rationality, so the behaviour of both parties will be affected by the influence of the other side of the game to make dynamic adjustments in order to obtain the maximum benefit. Participating subjects according to the vested interests of the continuous adjustment of strategy in pursuit of their own interests to improve, and ultimately achieve dynamic equilibrium strategy is called the evolutionary stability strategy (ESS). When F(x) = 0 and F(y) = 0, i.e., the rate of change of system strategy selection is zero, five equilibrium points of the dynamical system can be obtained, which are D1(C2+P-Fs)/(C2-aC2-Fs+aFs), (C1+E1)/(aE1+Fs+Gs-aFs-aGs), D2(0,0), D3(1,0), D4(0,1), D5 (1,1).

Asymmetric game only need to discuss the asymptotic stability of the pure strategy equilibrium can be, so do not consider D1, only need to discuss the four pure strategy Nash equilibrium points, respectively, D2 (0,0), D3 (1,0), D4 (0,1), D5 (1,1) will be pure strategy equilibrium points into the Jacobi matrix, find out the eigenvalues of each point corresponding to the Li Yapunov's first method can be known, when the eigenvalues are all When the eigenvalues are all negative, the equilibrium point is stable; when the eigenvalues are all positive, the equilibrium point is unstable; when the eigenvalues are both positive and negative, the equilibrium point is saddle point. The eigenvalues and stability analysis of each point are shown in Table 4.

Equilibrium point I D2: When Fs < C2 + P, the system has an evolutionary stable strategy (ESS) of (0, 0), at which time the system strategy is {collusion, laissez-faire}. To achieve this stable state, the infringement of the uncertainty brought about by the benefits of P to be large enough to make the infringed out of rational consideration to choose the laissez-faire strategy; or the cost of rights, the difficulty of rights is too high, even if the success of the rights of the infringement of compensation is not enough to make up for the cost of the infringed will choose to laissez-faire strategy, this time the platform does not have to worry about the risk of litigation, the natural tendency is to conspire to strategy, in order to obtain a greater return. At the same time, the lower the probability of successful defence, the system is more likely to converge on the stability point.

**Table 4. Stability analysis of equilibrium points for pure strategies.**

| balance point | eigenvalue (math.) | stability |
|---|---|---|
| D2(0,0) | - C1—E1 | When Fs < C2 + P, D2 is a stable point, otherwise it is unstable. |
|  | Fs—P—C2 |  |
| D3(1,0) | C1 + E1 | D3 is the point of instability or saddle point |
|  | aFs—aC2—P |  |
| D4(0,1) | C2 + P—Fs | D4 is stable when C2 + P < Fs and aE1 + Fs + Gs < E1 + C1 + aFs + aGs, otherwise it is unstable or saddle point |
|  | aE1—E1—C1 + Fs + Gs—aFs—aGs |  |
| D5(1,1) | P + aC2—aFs | D5 is stable when P+aC2 < aFs and E1+C1+aFs+aGs < aE1+Fs+Gs, otherwise it is unstable or saddle-pointed |
|  | C1 + E1—aE1—Fs—Gs + aFs + aGs |  |

Equilibrium point II D4: When C2+P<Fs and aE1+Fs+Gs<E1+C1+aFs+aGs, the system has an evolutionary stable strategy (ESS) of (0, 1), at which time the system strategy is {collusion, rights defence}. To achieve this stable state, the preconditions and equilibrium point I is the opposite, infringement brought about by the uncertainty of the benefits P to reduce or even negative benefits, or the reduction of the cost of rights and rights to increase the probability of success, the infringed person out of rational considerations will choose to defend the rights strategy. When the infringed chooses a rights defence strategy, even if the successful defence will result in the platform having to pay infringement damages and reputational damage, as long as the probability of successful defence is low enough, the platform will still take the risk of continuing to collude with the infringer. Or the additional cost of active regulation is too high, and the probability of regulatory failure is too high, which makes the platform reluctant to choose an active regulatory strategy. At this point, the higher the probability of regulatory failure, the lower the probability of successful rights defence, and the faster the system converges to that point of stability.

Equilibrium point III D5: When P+aC2 < aFs and E1+C1+aFs+aGs < aE1+Fs+Gs, the system has an evolutionary stabilisation strategy (ESS) of (1, 1), and at this point, the system strategy is {positive regulation, rights defence}. To achieve this stable state, the uncertain benefit P brought by the infringement should be reduced or even negative, or the cost of defending the right should be reduced and the probability of success of defending the right should be increased, and the infringed will choose the strategy of defending the right out of rational consideration. The conditions affecting the platform's strategy choice are the opposite of equilibrium two, where the platform, faced with high infringement damages and huge reputational losses, tends to favour an aggressive regulatory strategy. At this point, the probability of regulatory failure has a greater impact on the stability of the system, too high or too low a probability will lead to system instability, and the lower the probability of successful rights defence, the faster the system will converge to that stability point.

## 4 Simulation analysis

In order to verify the correctness of the above model analysis results, and at the same time more intuitively show the influence of different parameter value taking on the strategy evolution path and stable state of the platform and the infringed, this paper analyses by numerical simulation that the platform and the infringed, under the initial strategy probability taking the values of (0.2,0.8); (0.4,0.6); (0.6,0.4); and (0.8,0.2), respectively, the Stability of the system equilibrium strategy. According to the parameter settings and stability point constraints above, the parameters are assigned values, and the dynamic evolution of the three assignment scenarios from different initial values of x, y is shown in Figs 2–4. Scenario one (equilibrium point I D2): C1 = 120;E1 = 100;a = 0.2;C2 = 50;F = 300;G = 500;s = 0.5;P = 150; Scenario two (equilibrium point II D4): C1 = 120;E1 = 100;a = 0.2;C2 = 50;F = 200;G = 200;s = 0.5;P = -50; Scenario three (Equilibrium III D5): C1 = 120;E1 = 100;a = 0.2;C2 = 50;F = 300;G = 500; s = 0.5;P = -70.

### 4.1 Impact of uncertainty gains on the outcome of the evolutionary game

In order to explore the effect of the magnitude of P under positive gain on the subject's strategy choice, the simulation is conducted by setting P = 0, P = 100, and P = 200 respectively on the basis of the assignment of case one, and the results are shown in Figs 5 and 6. Meanwhile, in order to explore the effect of the magnitude of P under negative loss on the subject's strategy choice, the simulation is carried out by setting P = -20, P = -50, and P = -100 respectively on the basis of the assignment of case one, and the results are shown in Figs 7 and 8. In order to

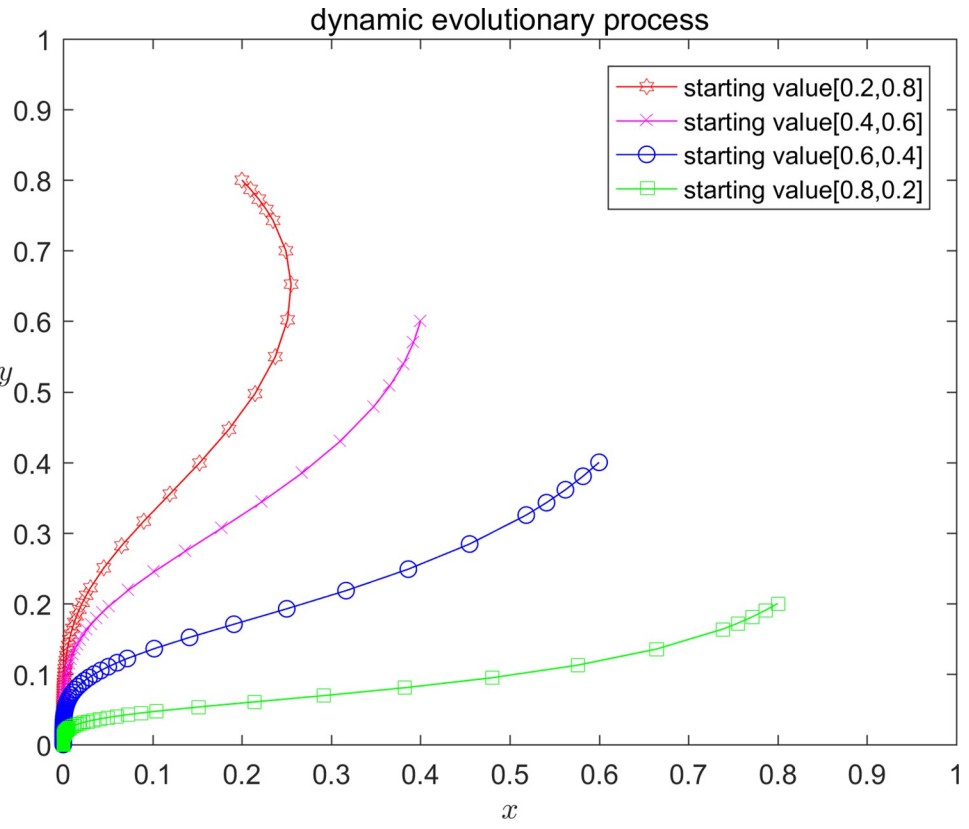

**Fig 2. Diagram of scenario 1 evolution.**

generalize the evolutionary results,the initial strategy probabilityvalues for both the platform and the infringed are set to a median value of 0.5.

As can be seen in Figs 5 and 6, when P = 0, the infringed's positive gain from being infringed is low, so the choice of strategy tends to be the rights defence strategy, at this time the platform is afraid of the infringed's success in rights defence by the legal penalties as well as the loss of reputation, the platform also tends to actively regulate the strategy. When P = 100, the infringed's strategy choice is not stable, may defend the right or may let go, this is because with the positive benefit of being infringed increases, the infringed defends the right of the uncertain benefit and choose to let go of the strategy to bring the positive benefit of being infringed on the gap between the smaller, so the infringed's strategy choice is extremely unstable. Platforms, on the other hand, gradually converge towards collusive strategies. When P = 200, the infringed tends to favour the laissez-faire strategy due to the rationality assumption, because the infringed obtains more positive gains when laissez-faire, and the platform chooses to collude with the infringed at this point because there is no more risk of being sued.

As can be seen in Figs 7 and 8, When P is a positive benefit, and P is greater than the infringement damages minus costs from the infringed's rights defence, the system strategy will stabilise at {collusion, indulgence}. This phenomenon is common in today's network environment, the infringed knows that his copyright has been infringed, but due to the difficulty of defending the right, the difficulty of evidence, the cost is greater than the benefit, have no choice but to choose to let it go; or because of the infringing work for the infringed to bring a positive uncertainty of the benefit of the P is much greater than the infringement of copyright infringement compensation can be obtained by the infringed, the infringed is based on

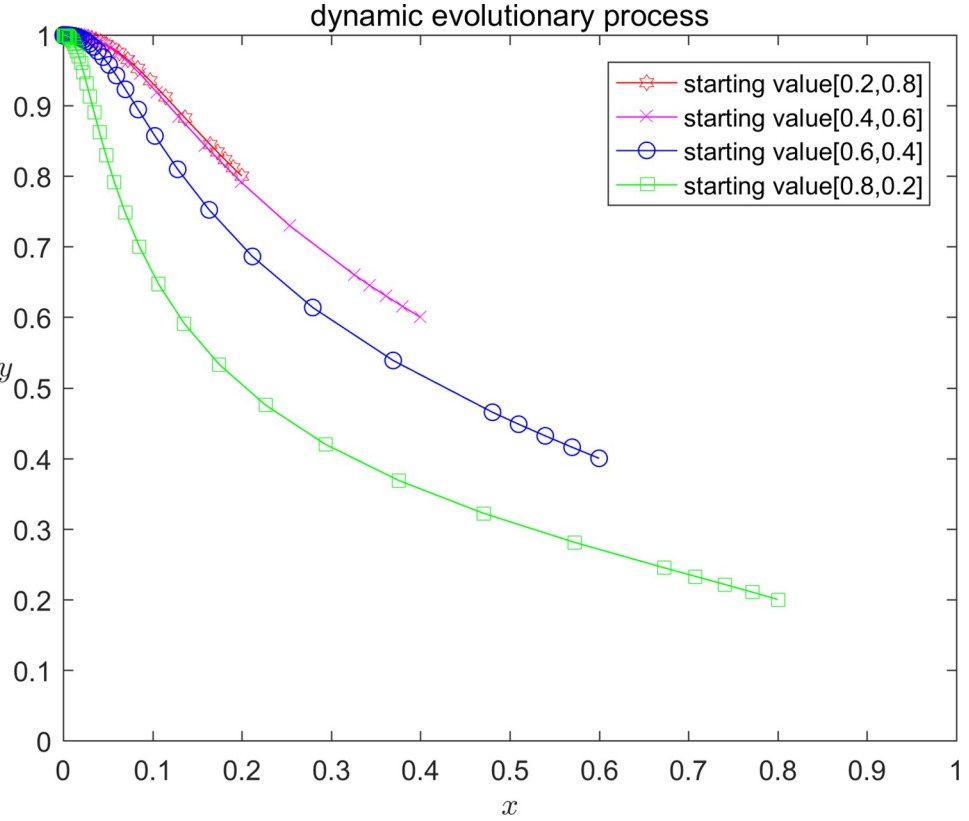

**Fig 3. Diagram of scenario 2 evolution.**

rational considerations, the choice of the strategy of laissez-faire. At this point for the platform, there is no longer the risk of the defendant, the natural choice of their own more favourable collusion strategy. But such a systematic strategy for the platform industry, the network environment, and even the social climate is extremely harmful.

When P is a negative benefit, the system strategy will be stabilised at {active regulation, defend rights}. The infringed brings negative revenue after being infringed, and in order to safeguard his own interests, he chooses the right defence strategy, so as to reduce the negative revenue brought by being infringed, thus promoting the platform to choose the active regulation strategy in order not to bear the risk of infringement. Such a systematic strategy is conducive to the ecological development of the platform industry and promotes the generation of more quality original content. However, the premise of the infringed's choice of rights protection strategy is the high probability of success, low cost and low difficulty of rights protection, which requires the introduction of corresponding legal regulations by the government legal department, under the guidance of the normative value of safeguarding order, to provide direction and convenience for the infringed's rights protection.

## 4.2 Impact of infringement compensation on the outcome of the evolutionary game after successful

**Rights defence.** In order to explore the effect of the size of f on the subject's strategy choice when p is a positive gain, each parameter is assigned a value ($C1 = 120; E1 = 100; a = 0.2; C2 = 50; G = 300; s = 0.5; P = 50$), and the simulation is performed by setting F = 100, F = 200,

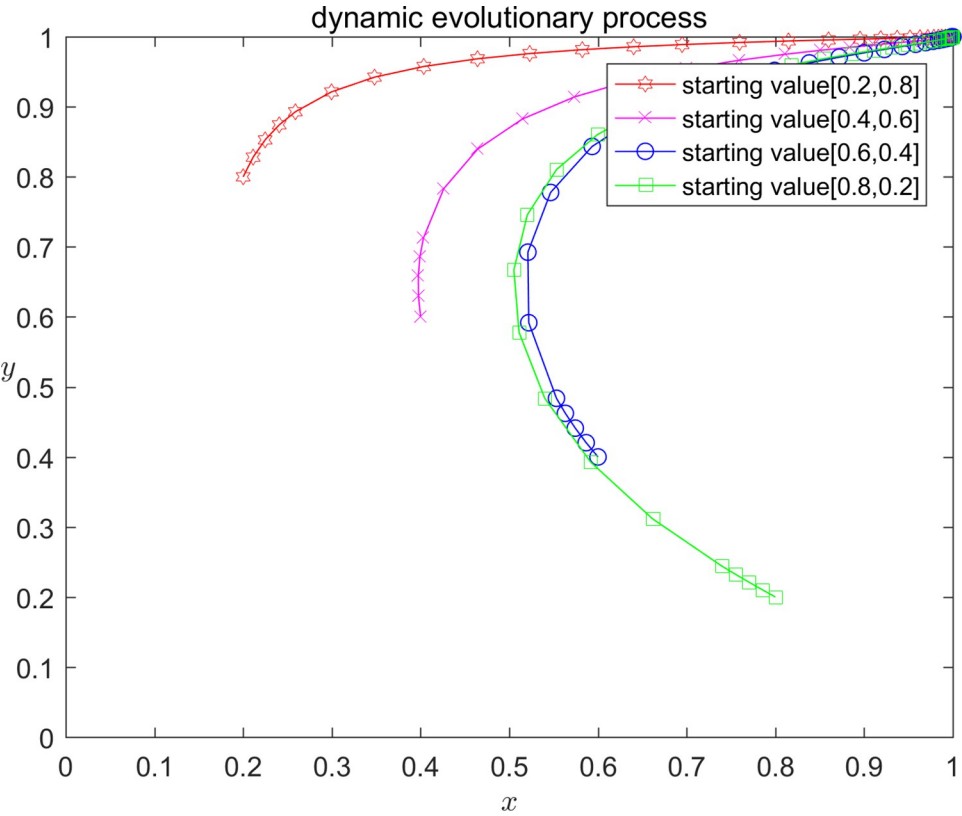

**Fig 4. Diagram of scenario 3 evolution.**

and F = 600, respectively, and the results are shown in Figs 9 and 10. Meanwhile, in order to explore the effect of the size of F on the subject strategy choice when P is negative gain/loss, each parameter is assigned a value (C1 = 120;E1 = 100;a = 0.2;C2 = 50;G = 500;s = 0.5;P = -50) and the simulation is carried out by setting F = 20, F = 100, F = 200, respectively, and the results are shown in Figs 11 and 12.

As can be seen from Figs 9 and 10, F represents the infringement compensation from the platform to the infringed after the successful defense of the right. P is a positive benefit and F = 100 or lower, at this time, the infringed gets a lower infringement compensation after the successful defense of the right, and the platform needs to pay lower infringement compensation, the platform believes that the additional benefit obtained from collusion is greater than the infringement compensation paid, and the infringement compensation after the successful defense of the right of the infringed is insufficient to cover the cost of the defense, the system strategy is {Collusion}, and F = 100 or lower. So the system strategy at this time is {collusion, indulgence}. With the increase of F, it means that the government departments pay more attention to such cases, as well as the focus of public opinion, the penalty for such cases increases, and the system strategy is also approaching to {active regulation, rights defense}. It can be found that even when F = 600 the platform has converged to the active regulation strategy, the infringed's strategy choice is still in an unstable state. This is because the infringed is already positive based on the existing revenue, while the compensation after successful defense is uncertain, and even the cost paid for defense may not be recovered. So the infringed's choice is unstable.As can be seen from Figs 11 and 12, when P is a negative return, the system strategy stabilizes at {active regulation, rights defense} requiring lower infringement compensation

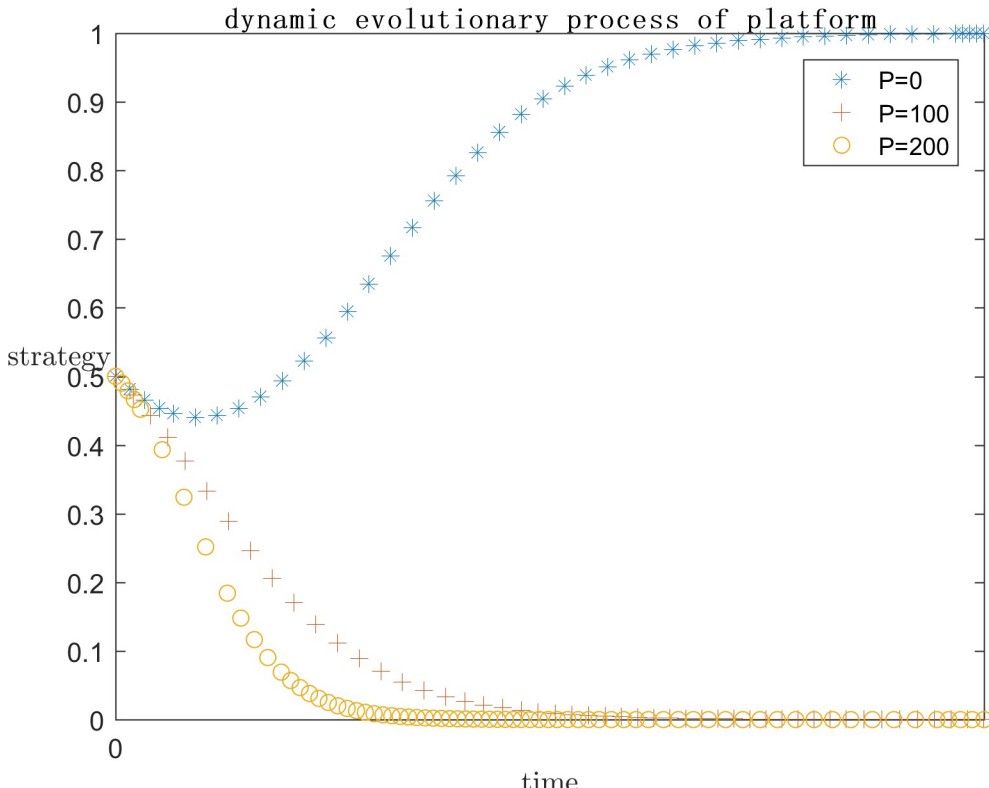

**Fig 5. Positive P impact on platform strategy.**

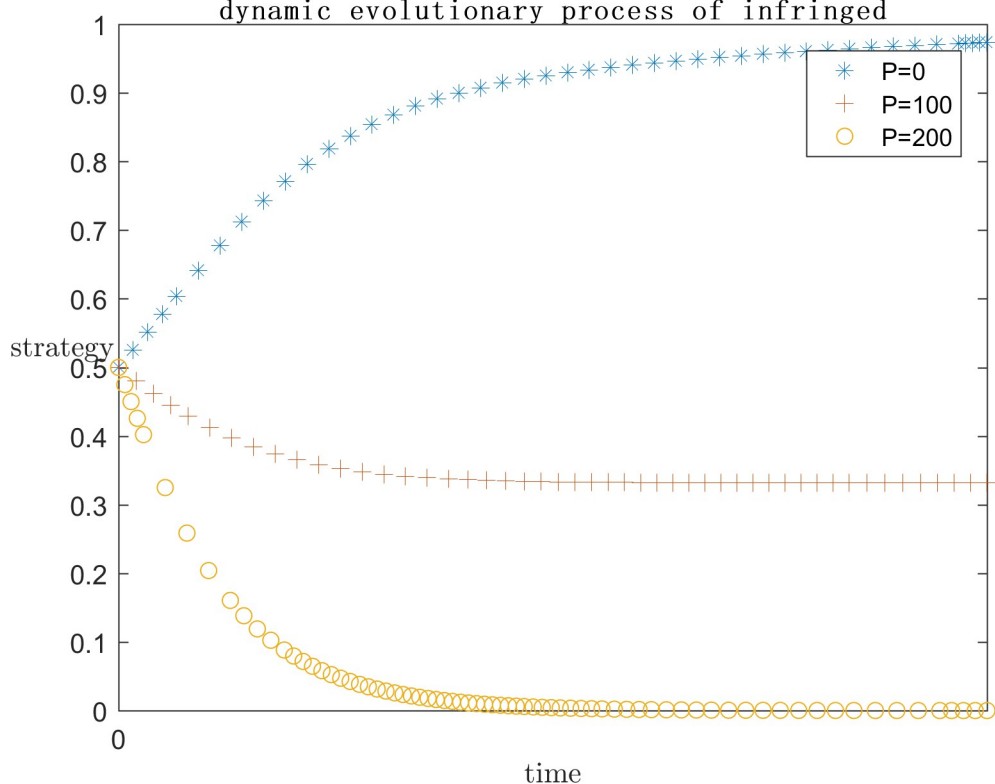

**Fig 6. Positive P impact on infringed strategy.**

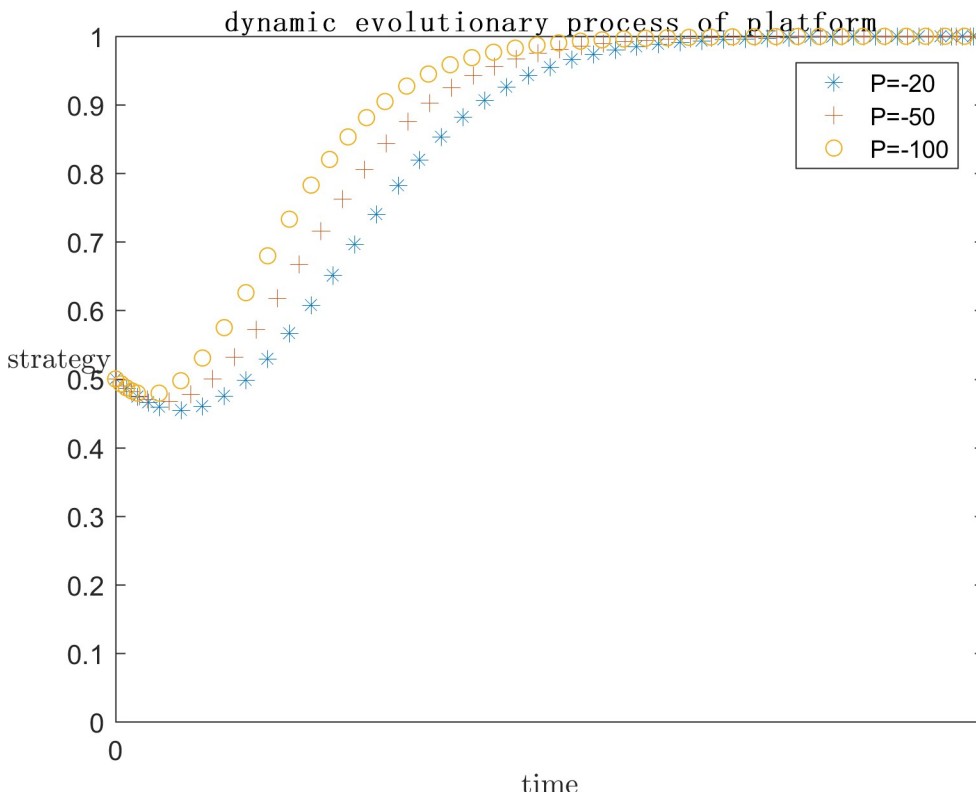

**Fig 7. Negative P impact on platform strategy.**

after successful rights defense than when it is a positive return, which means that the platform and the infringed are more willing to actively regulate and defend their rights.

As the set value of F becomes larger and larger, i.e., the infringement compensation becomes higher and higher, the infringed's infringement compensation after successfully defending his rights becomes larger and larger, and the platform collusion pays more and more, the system strategy will gradually stabilize at {positive regulation, defending rights}. When P is a positive return, F needs to reach 600 for the platform strategy to stabilize at active regulation. And when F is only 200 for negative returns, the platform strategy will stabilize in active regulation. Therefore, in the legal regulation of indirect copyright infringement by algorithms, the government legal department should increase the penalty for the platform and the infringement compensation for the infringed. However, in the case that P is positive gain, even if F = 600 is set, which is far more than the reality, the infringed's strategy choice is only tends to defend the right but still not stabilized in the defense strategy. Visible, want to make the system strategy stabilized in the {positive regulation, rights protection} only rely on legal policy to improve the infringement penalty infringement compensation is far from enough, but also need to ensure that the probability of success of rights protection, to encourage the infringed to reasonably maintain their own copyright.

## 4.3 Impact of reputational loss of platforms after successful rights defence on the outcome of the evolutionary game

In order to explore the effect of the size of G on the subject's strategy choice when P is a positive return, based on the assignment of (C1 = 120;E1 = 100;a = 0.2;C2 = 50;F = 500;s = 0.5;

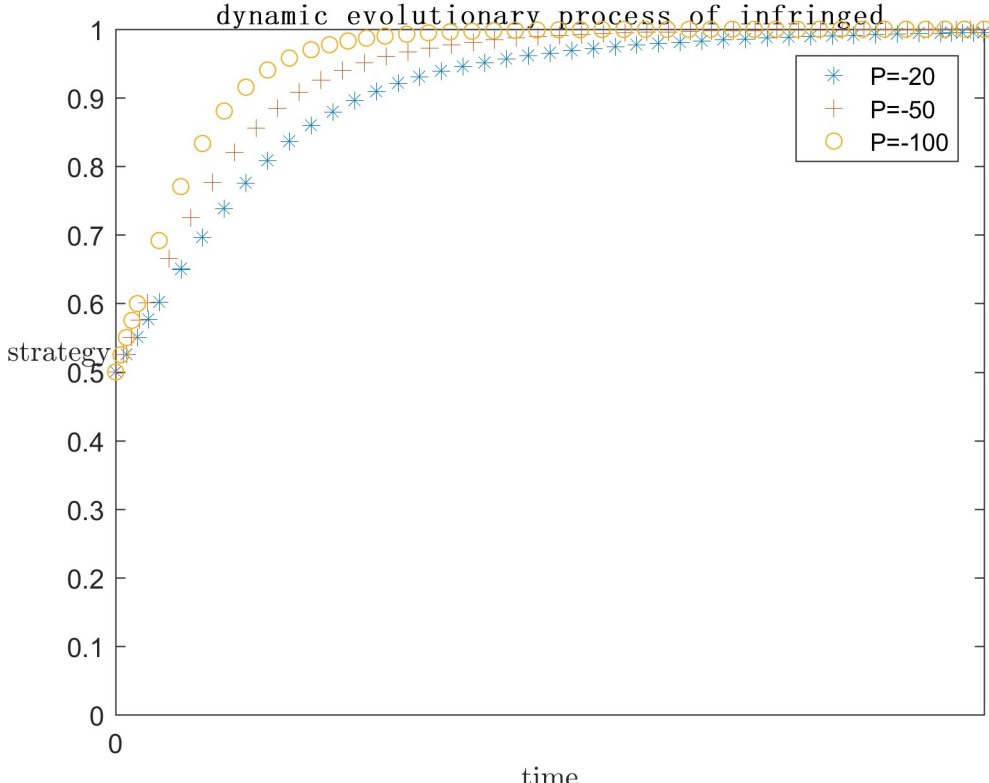

**Fig 8. Negative P impact on infringed strategy.**

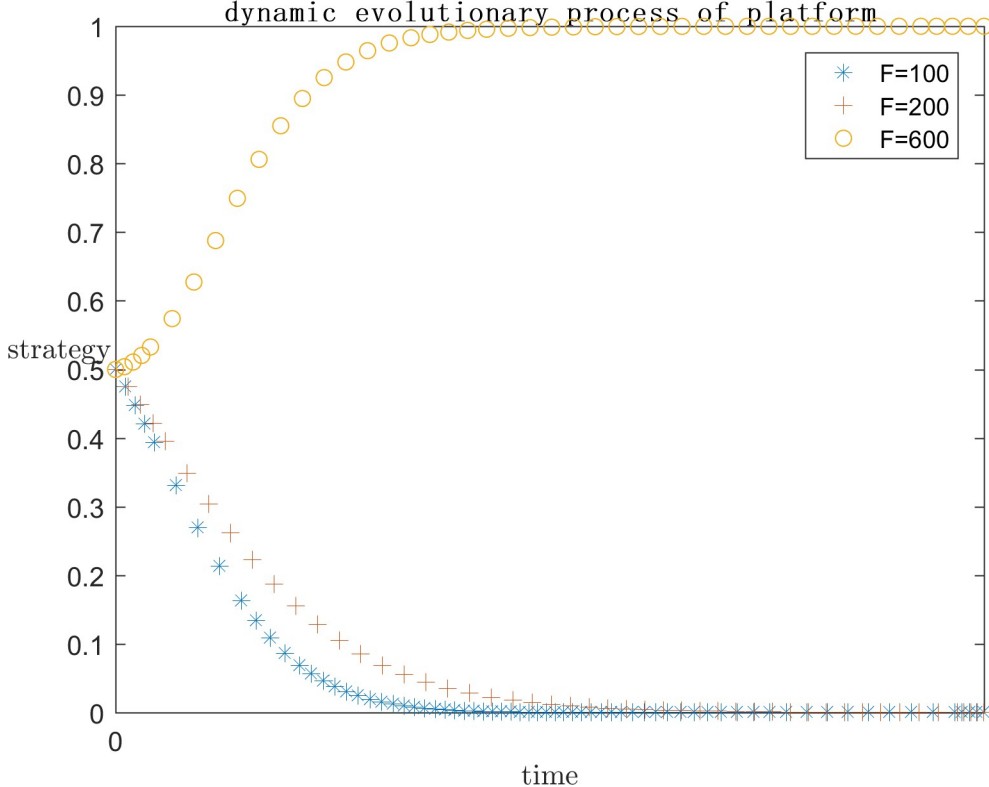

**Fig 9. Impact of F on platform strategy in the positive P case.**

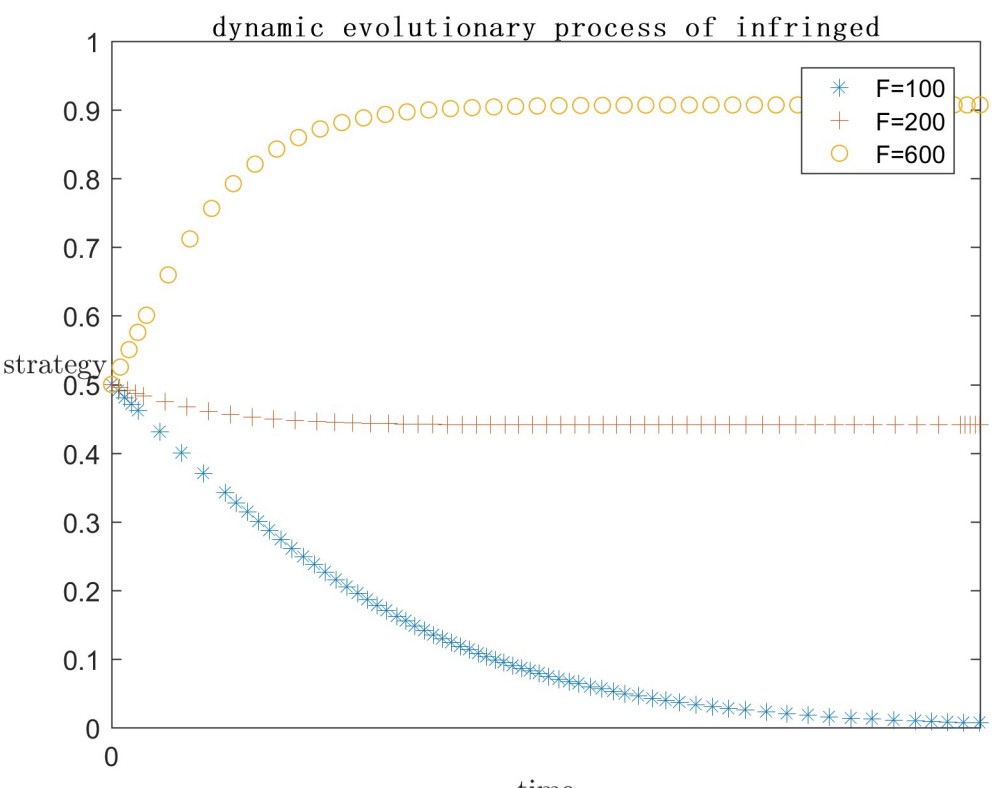

**Fig 10. Impact of F on infringed strategy in the positive P case.**

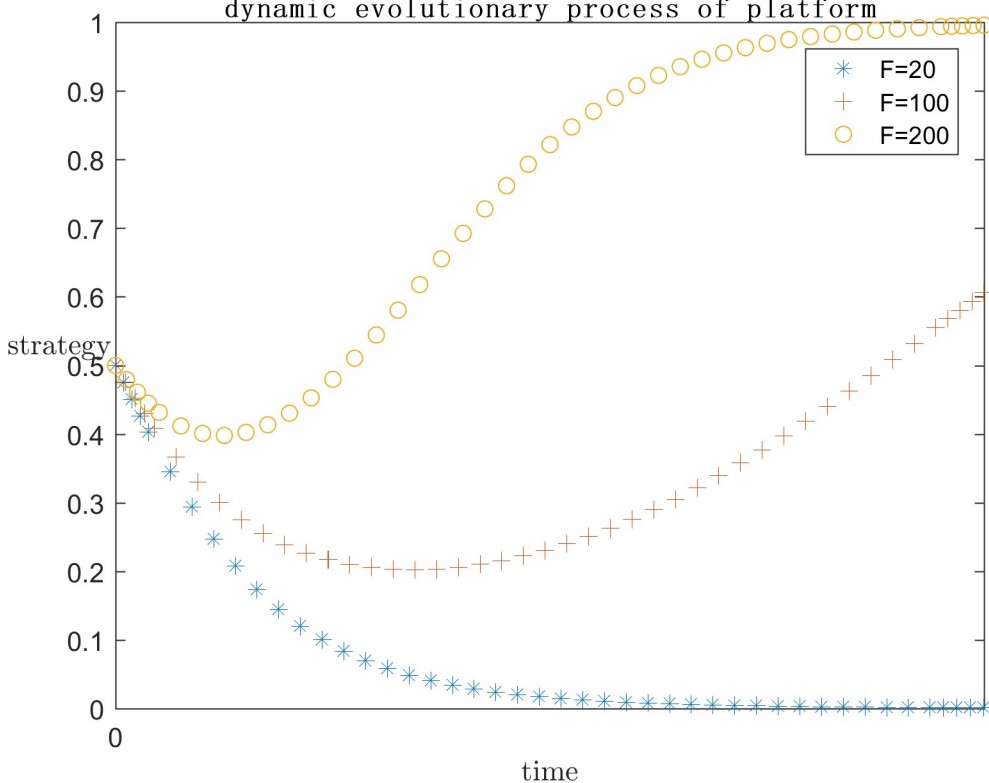

**Fig 11. Impact of F on platform strategy in the negative P case.**

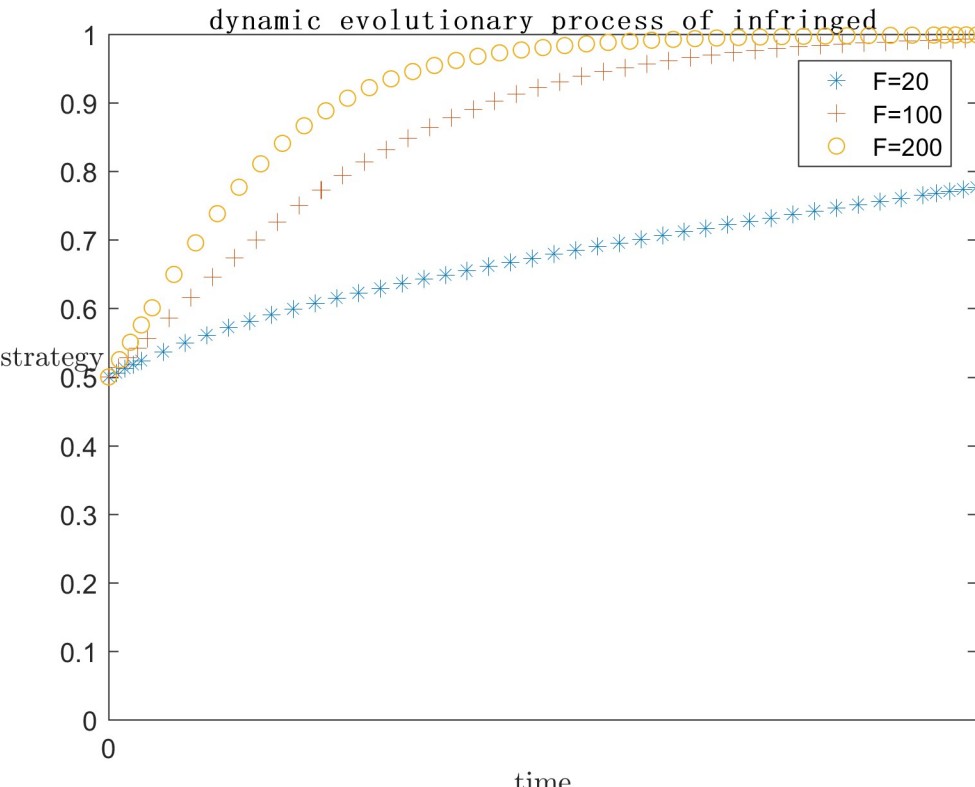

**Fig 12. Impact of F on infringed strategy in the negative P case.**

P = 50), the simulation is carried out by setting G = 100, G = 300, and G = 500 respectively, and the results are shown in Figs 13 and 14. At the same time, in order to explore the effect of the size of G on the subject's strategy choice when P is a negative return, each parameter is assigned a value (C1 = 120; E1 = 100; a = 0.2; C2 = 100; F = 200; s = 0.5; P = -50), and the simulation is performed by setting G = 100, G = 300, and G = 500 respectively, and the results are shown in Figs 15 and 16.

As can be seen from Figs 13–16, G gets progressively larger, i.e., the reputational damage inflicted on the platform by the successful defense of the infringed's rights becomes larger and larger, the platform gradually stabilizes its convergence to an active regulatory strategy. Comparing the evolution process, it can be found that as G increases, the platform tends to positive regulatory strategy faster and faster. When P is a positive gain, the choice of strategy for the infringed shows an inverse proportional function. This is the same as the situation when the infringement compensation is raised, where platforms are afraid of bearing the reputational damage of a successful defense and thus have a higher probability of choosing an aggressive regulatory strategy. So the infringed, even if he knows he is infringed, chooses to wait for the platform's regulatory measures to restrict the infringing content instead of directly choosing a rights defense strategy that requires additional costs. When P is a negative benefit, the change of G has almost no effect on the infringed's strategy choice.

As can be seen from Figs 13–16, it can be seen that when p is a positive benefit, the change of G makes the platform's strategy stabilize more quickly to positive regulation compared to when P is a negative benefit. Therefore, in view of the current network status quo, when P is more likely to be positive, it is more important to raise public awareness of copyright

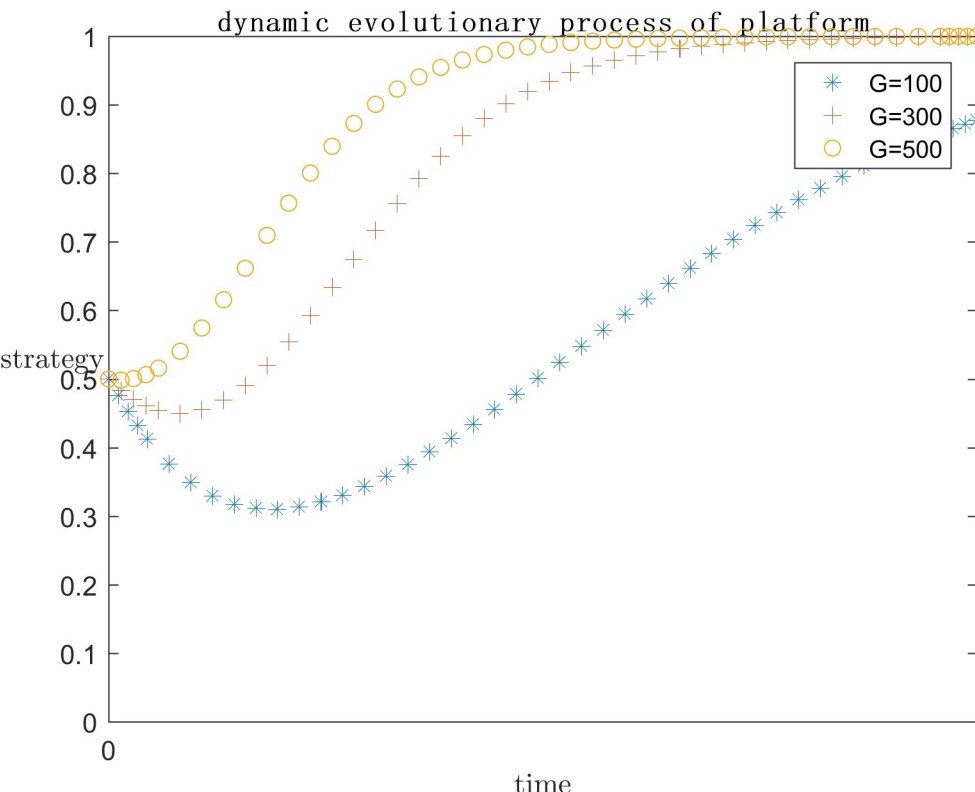

**Fig 13. Impact of G on platform strategy in the positive P case.**

protection, so as to protect the ecological environment of the platform industry by making the reputational damage suffered by platforms' collusive strategies more serious.

## 5 Recommendations for countermeasures

The arrival of the Web 2.0 era has brought about the rapid development of network platforms, prompting the problem of user-generated content to emerge gradually, especially the problem of indirect copyright infringement by algorithms is particularly prominent, and UGC platforms, in response to the country's strategic decision to create a clear cyberspace in the context of the need for effective governance strategies to promote the healthy development of the UGC platform industry. Previous studies have not yet carried out effective analysis and mathematical modelling of the inter-subjective relationship of interests in infringement incidents. In this paper, for the governance of algorithmic infringement on UGC platforms, we constructed a two-party evolutionary game model of "platform—infringed", set up a mechanism of "collusion between the platform and infringer", and built a two-party evolutionary game model of "platform—infringed", which is centred on the different uncertain benefits of the infringed, the compensation for infringement, and the compensation for sound infringement suffered by the platform after the infringement. The study also analyses the interaction mechanism between the two game players, and analyses the stability of the system and the evolution path under different conditions. It is found that there are three equilibrium points in the game between the UGC platform and the infringed, and the system should satisfy the equilibrium point three (1, 1) for a healthy and lasting development, and the strategy should be stable at {positive regulation, rights protection}, which needs to satisfy the conditions that $P+aC2<aFs$

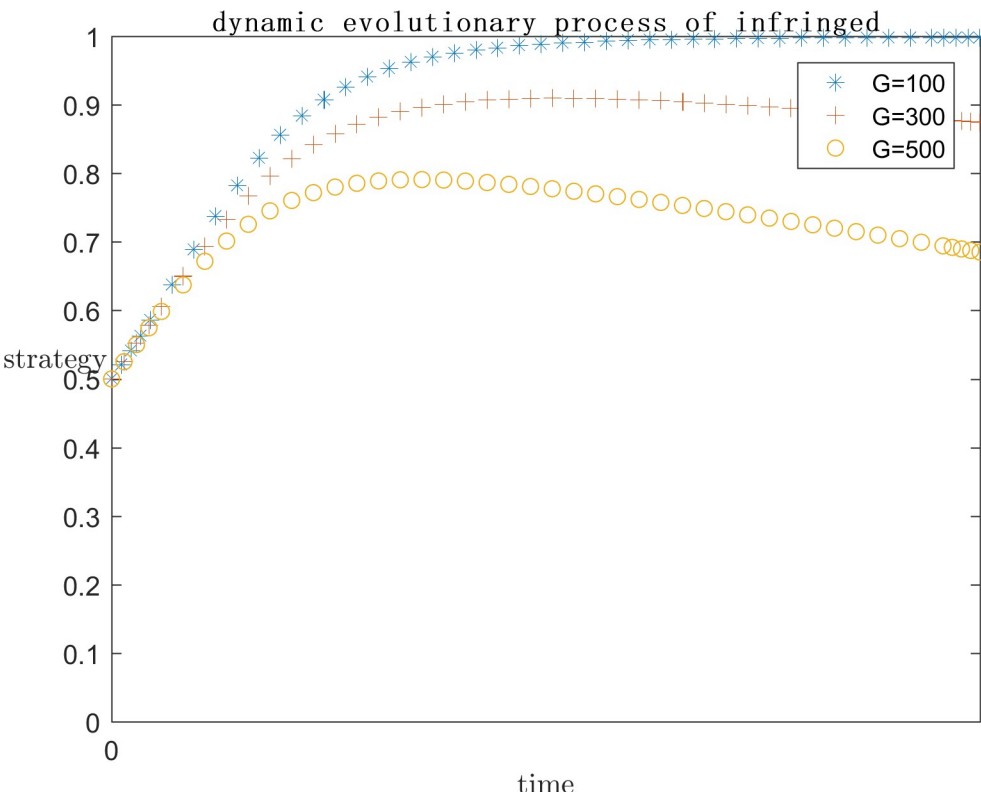

**Fig 14. Impact of G on infringed strategy in the positive P case.**

and $E1+C1+aFs+aGs<aE1+Fs+Gs$. Combined with the results of the simulation, we propose the following recommendations at the level of the content publisher, the platform and the governmental law. governmental legal level, the following suggestions are made:

Content publishers should consciously raise their awareness of copyright protection, take practical action to jointly protect every intellectual achievement, and create a healthy and favourable online environment for China's content industry. Nowadays, the uncertain gain brought by infringement is often positive, and even far more than the compensation brought by the success of the right to defend, which is a higher requirement for each copyright owner, the need to give up the uncertain gain of the moment", in the face of infringement to do the first time to defend the right to sue. Otherwise, the infringed is no longer afraid of the risk of copyright infringement, and will simply copy and steal others' original works, which will eventually drive out the good money and lead to "no one can copy". Content publishers in reference to the works of others, should be under reasonable and lawful regulations. Secondly, they should avoid directly copying or carrying other people's original works. But the most important thing is still to adhere to the original, and strive for innovation. At the same time, they should add obvious signs and relevant rights instructions on their original works, and pay attention to collecting evidence when their original works are infringed upon, so as to facilitate the subsequent defence of rights.

If the platform wants to develop sustainably, it should not lose its sense of social responsibility for the sake of profit, and should consciously establish an effective regulatory mechanism to achieve platform self-purification. First of all, we should form a technical means of prevention, can use the content identification system, increase the user-generated content for artificial or algorithmic audit, and for the possible infringement of content to reduce the strength of the

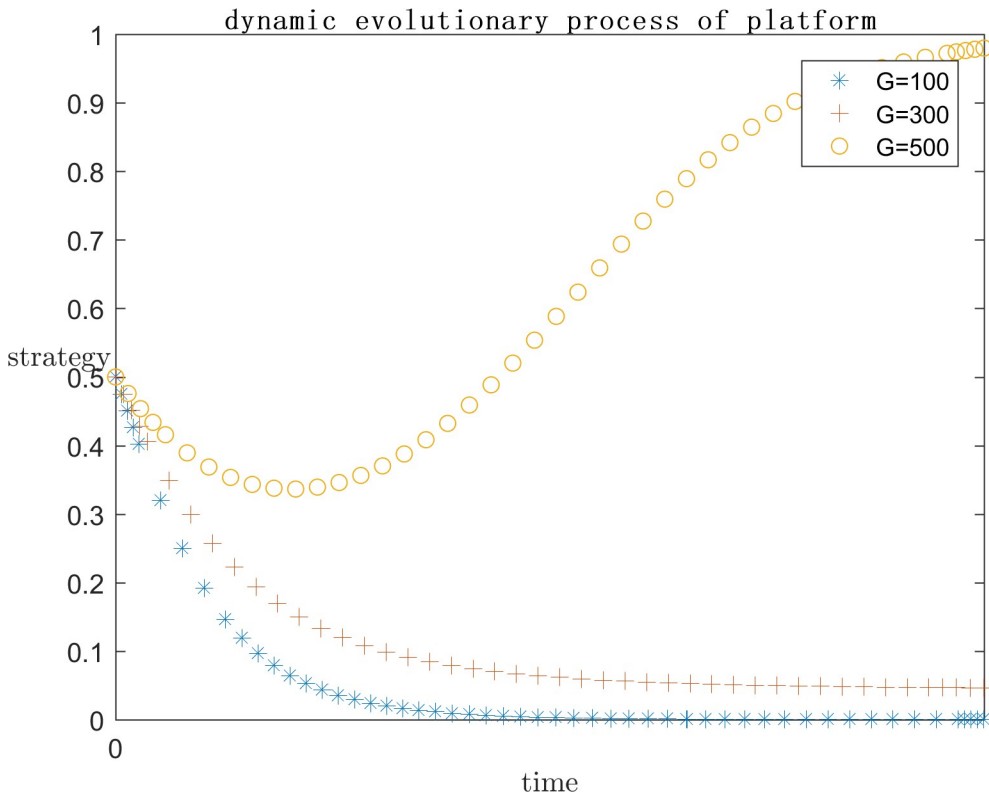

**Fig 15. Impact of G on platform strategy in the negative P case.**

recommendation algorithm, for its account to issue a notice warning, to limit the allocation of the account traffic, in order to reduce the risk of indirect infringement of the algorithm. Secondly, we can promote the "micro-copyright" mechanism and establish a new model of winwin cooperation, whereby the platform obtains the authorisation of most copyright holders through centralised licensing in a one-stop manner, so that all users within the platform can freely use the relevant works within the platform, which not only expands the influence and visibility of the works of copyright holders, but also reduces the cost of legal use of works by content creators, which is conducive to stimulating the development of the content industry. This not only expands the influence and popularity of the copyright owner's works, but also reduces the cost of legal use of works for content creators, which is conducive to stimulating the enthusiasm of content creators and the output of excellent original works. Finally, UGC platforms should actively carry out user support programmes, including providing direct economic incentives, professional training and guidance for original creators, guiding content creators to produce content according to the platform's preset norms and processes, building collective values within the platform, and realising the platform's ecological co-creation.

China's existing legal norms for algorithmic infringement provisions are still not clear and specific, should improve the legal mechanism to protect the legitimate rights and interests of works. The first to implement the "punitive damages system", significantly increase the cost of infringement is undoubtedly an effective means of governance recommended algorithm indirect infringement. Improve the cost of infringement, algorithmic indirect infringement will be effectively curbed, so that more original creators of the legitimate rights and interests of legal protection. The second is to regulate the obligation of attention of the platform and clarify the management responsibility of the platform, because the platform is not only the operator of

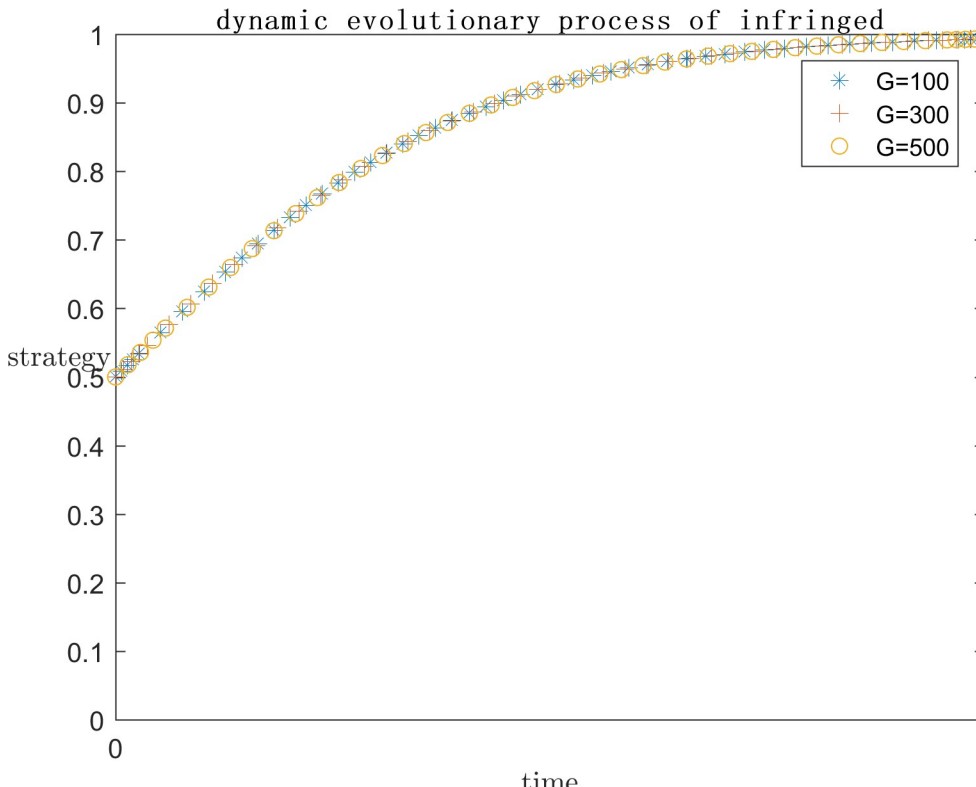

**Fig 16. Impact of G on infringed strategy in the negative P case.**

the cyberspace, but also the manager of this particular cyberspace, so the platform has the obligation and responsibility to review and manage the content in the cyberspace. Thirdly, the existing law does not compel the platform to provide the real name information of the direct infringer, which is also the most direct reason for the infringed person's difficulty in defending his rights. Therefore, when an infringement incident occurs, the judicial authorities should help the infringed to find the direct infringer, or require the platform to provide the real name information of the direct infringer, so as to reduce the difficulty of defending the rights, encourage the infringed to actively defend the rights, and effectively protect the legitimate rights and interests of the copyright owner, and maintain the balance of interests of all parties.

This study reveals the two-party game between UGC platforms and infringerd in algorithmic infringement incidents. The greatest contribution of this study is the application of the two-party evolutionary game approach to algorithmic infringement and provides various insights into infringement governance. It should be noted that this study has various limitations that open the way for further research. First, direct infringer were not added to the model in order to simplify the model. However, the strategic choices of direct infringer also affect the stability of the system to some extent in actual infringement events. Second, the variable design of this study is based on the assumption of common scenarios, but there is no specific variable data collected from real cases, and inevitably there are other variables not considered, and the generalizability of the research conclusions needs to be strengthened. Third, our study only examines the indirect infringement of algorithms occurring on UGC platforms. However, for different online platforms, such as social platforms and e-commerce platforms, the penalties and benefits after the occurrence of algorithmic indirect infringement incidents are not exactly the same as those for UGC platforms. Future research can join the

direct infringer to form a three-way game, and also explore how the strategy choices generated by algorithmic indirect infringement incidents occurring on different online platforms are different.

## Supporting information

**S1 Data. This is the numerical simulation example data.**
(XLSX)

## Author Contributions

**Data curation:** Yuxuan Shen.

**Formal analysis:** Yuxuan Shen.

**Funding acquisition:** Jiangang Liu.

**Methodology:** Lanlan Zhou.

**Project administration:** Lanlan Zhou.

**Supervision:** Jiangang Liu.

**Writing – original draft:** Yuxuan Shen.

**Writing – review & editing:** Jiangang Liu.

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
