## [Decision Letter · Decision Letter 0]

27 Nov 2023

PONE-D-23-30742Evolutionary game analysis of indirect copyright infringement by algorithms from the perspective of collusion between UGC platforms and infringersPLOS ONE

Dear Dr. Shen,

Thank you for submitting your manuscript to PLOS ONE. After careful consideration, we feel that it has merit but does not fully meet PLOS ONE’s publication criteria as it currently stands. Therefore, we invite you to submit a revised version of the manuscript that addresses the points raised during the review process.

 **Academic Editor Comments: **

First, the study design is not clearly explained. It is unclear how the participants were selected and randomized, and the authors do not provide any information on the blinding of participants or investigators. This could potentially affect the validity of the study results.

We look forward to receiving your revised manuscript.

Kind regards,

Mudassir Khan, Ph.D

Academic Editor

PLOS ONE

Journal Requirements:

"National Social Science Fund of China“Research on the Governance Mechanism of Online Platform Enterprises from the Perspective of Quasi-Public Goods”(20BGLO96);Social Science Foundation of Jiangsu Province“Research on Risk Generation Mechanism and Countermeasures for Prevention and Control of Network Platform Enterprises from the Perspective of Life Cycle "(19GLBO15);"

Additional Editor Comments:

First, the study design is not clearly explained. It is unclear how the participants were selected and randomized, and the authors do not provide any information on the blinding of participants or investigators. This could potentially affect the validity of the study results.

Dear Authors,

You are advised to work on Editor/reviewers comments.

Reviewers' comments:

Reviewer's Responses to Questions

**Comments to the Author**

1. Is the manuscript technically sound, and do the data support the conclusions?

Reviewer #1: Partly

Reviewer #2: Yes

2. Has the statistical analysis been performed appropriately and rigorously? 

Reviewer #1: I Don't Know

Reviewer #2: Yes

3. Have the authors made all data underlying the findings in their manuscript fully available?

Reviewer #1: No

Reviewer #2: Yes

4. Is the manuscript presented in an intelligible fashion and written in standard English?

Reviewer #1: Yes

Reviewer #2: Yes

5. Review Comments to the Author

Reviewer #1: Title: Evolutionary game analysis of indirect copyright infringement by algorithms from the

the perspective of collusion between UGC platforms and infringers

Review

# Comment to the author:

The article focuses on the challenges and dynamics related to copyright infringement on user-generated content (UGC) platforms and the development of strategies and measures to address these issues, emphasising factors affecting infringement governance decisions and promoting stability and legality in the UGC ecosystem.

#Introduction:

The introduction reflects the discussion's content and meets the publication standard.

# Model construction

The model description is understandable, using symbols to represent different variables. However, in #2.3 Model assumptions and benefits matrix:

Such as : 11 = [−1 + (1 − − )] + (1 − )(−1). The use of subscript would have been preferred for V11. i.e., V_11.

# Impact of Infringement Compensation on the Outcome of Evolutionary Games after Successful Rights Defense:

Since simulated data are employed in describing the outcome of Evolutionary Games used in modelling infringement, more statistical output data need to be displayed

Try to compute using the simulated data with the algorithm and show the outcome of at least 2 – 3 instances. This will enhance understanding of the propose algorithm.

#Figures - Labelling the graphs using English would be preferred for the reader to understand the pictorial view representing the outcome of the evolutionary game.

# Countermeasures and recommendations

The recommendations reflect the importance of the algorithm proposed and demonstrated using a simulated dataset in the article. However, much is expected for the author to suggest future research based on a wealth of understanding gained in this domain while solving this problem.

Reviewer #2: 1- Modify reference 0 to reference 1 in the following statement: UGC (User Generated Content) platform refers to various forms of media and creative works created by users

through the Internet and its technology.0

2-Arranging the references mentioned in the introduction according to the year of publication (from oldest to newest) for the research review mentioned in this section

3- Adding a paragraph at the end of the introduction that explains the general structure of the research by clarifying each section

4- Standardize table content formats

5- Section 2, in all its paragraphs, is devoid of reference to any reference

6- Explaining the proposed algorithm in clear steps or through a flow chart that summarizes what has been explained

7-The proposed future work has not been clarified to provide a future outlook for researchers in this field

8-Standardize the format of references

6. PLOS authors have the option to publish the peer review history of their article (what does this mean?). If published, this will include your full peer review and any attached files.

Reviewer #1: **Yes: **Adebayo, Paul Olujide

Reviewer #2: No

---

## [Author Response · Author response to Decision Letter 0]

28 Dec 2023

Reviewer#1:

Response to#2.3 Model assumptions and benefits matrix:

Such as : 11 = [−1 + (1 − − )] + (1 − )(−1). The use of subscript would have been preferred for V11. i.e., V_11.

Response:We appreciate the heads-up and have made changes in the text(line183-188).

Response to#Impact of Infringement Compensation on the Outcome of Evolutionary Games after Successful Rights Defense:Since simulated data are employed in describing the outcome of Evolutionary Games used in modelling infringement, more statistical output data need to be displayed.Try to compute using the simulated data with the algorithm and show the outcome of at least 2 – 3 instances. This will enhance understanding of the propose algorithm.

Response:We have listened carefully to your comments. And we have added four cases of arithmetic evolution and added Figures 7 and 8 for easier understanding.(line:300-354).

Response to#Figures - Labelling the graphs using English would be preferred for the reader to understand the pictorial view representing the outcome of the evolutionary game.

Response:Your reminder is much appreciated and we have changed all the graphs in the text to English.

Response to#Countermeasures and recommendations

The recommendations reflect the importance of the algorithm proposed and demonstrated using a simulated dataset in the article. However, much is expected for the author to suggest future research based on a wealth of understanding gained in this domain while solving this problem.

Response:We greatly appreciate your suggestions and have included suggestions for future research at the end of the article.(line:448-463)

Reviewer#2:

Response to#2: 1- Modify reference 0 to reference 1 in the following statement: UGC (User Generated Content) platform refers to various forms of media and creative works created by users

through the Internet and its technology.0

Response:We appreciate your suggestions and have completed the changes in the text.

Response to 2-Arranging the references mentioned in the introduction according to the year of publication (from oldest to newest) for the research review mentioned in this section

Response:Sorry, we didn't quite understand what you meant. But we have arranged the references from old to new and put them at the end of this letter. We hope it will meet your requirements.

Response to3-Adding a paragraph at the end of the introduction that explains the general structure of the research by clarifying each section.4- Standardize table content formats

Response:Thanks for the heads up, we've included a paragraph at the end of the introduction as an explanation(line:107-110). Harmonization of the formatting in the tables has also been completed.

Response to 5- Section 2, in all its paragraphs, is devoid of reference to any reference. 6- Explaining the proposed algorithm in clear steps or through a flow chart that summarizes what has been explained

Response:We are grateful for the suggestion. To be more clear and in accordance with the reviewer concerns, we have added a brief description as follows: "Background information on evolutionary games is included in the text(line:111-118), explaining why the use of evolutionary games was chosen to describe this type of infringement.The strategic choices of the participating subjects are also depicted in the form of a flowchart(line:160-169).”

Response to 7-The proposed future work has not been clarified to provide a future outlook for researchers in this field.

Response:We greatly appreciate your suggestions and have included suggestions for future research at the end of the article.(line:448-463)

Response to 8-Standardize the format of references

Response:We really apologize for not standardizing the format of references and have made changes.

---

## [Decision Letter · Decision Letter 1]

29 Feb 2024

PONE-D-23-30742R1An Evolutionary Game Analysis of Algorithmic Indirect Copyright Infringement from the Perspective of Collusion between UGC Platforms and direct InfringersPLOS ONE

Dear Dr. Shen,

Thank you for submitting your manuscript to PLOS ONE. After careful consideration, we feel that it has merit but does not fully meet PLOS ONE’s publication criteria as it currently stands. Therefore, we invite you to submit a revised version of the manuscript that addresses the points raised during the review process.

We look forward to receiving your revised manuscript.

Kind regards,

Mudassir Khan, Ph.D

Academic Editor

PLOS ONE

Additional Editor Comments:

Dear Authors,

The introduction provides a good, generalized background of the topic that quickly gives the reader an appreciation of the wide range of applications for this technology. This section helpfully explains the motivation for the research to current and potential funding agencies. However, to make the motivation clearer and to differentiate the paper some more from other applied papers, the author may wish to provide another sentence giving examples of some of the applications of this technology, along with appropriate references.

The objective is clearly defined in the last paragraph of introduction section and clearly mentions the flow of the paper. The experimental apparatus is quite standard, and is appropriate for the study, especially given that the focus of the paper is to develop a privacy preserving platform for industrial applications.

I don’t think any additional experiments are necessary to validate the results presented here, because the results themselves are not what is important; it is the technique used to obtain these results that is important. One exception to this reasoning would be if the author could demonstrate that the results obtained using the present method are consistent with results obtained using a different technique.

There are several instances where assertions are made that are not substantiated with references. These have been noted in the appropriate sections of this paper. it would be better if you provide references to these assertions as this will make your work more concrete.

Author may wish to elaborate the problem formulation section and add some more details which will make the paper look more vital. I do not think any additional graphics are necessary. The author may also wish to give a more detailed discussion on blockchain and machine learning implementation.

The author may wish to mention why it is important to leverage blockchain to explain the motivation for his choice of specimens and accompany this with some references to other studies that demonstrate this importance.

You're requested to work on reviewers comments as well.

Reviewers' comments:

Reviewer's Responses to Questions

**Comments to the Author**

1. If the authors have adequately addressed your comments raised in a previous round of review and you feel that this manuscript is now acceptable for publication, you may indicate that here to bypass the “Comments to the Author” section, enter your conflict of interest statement in the “Confidential to Editor” section, and submit your "Accept" recommendation.

Reviewer #2: All comments have been addressed

Reviewer #3: All comments have been addressed

2. Is the manuscript technically sound, and do the data support the conclusions?

Reviewer #2: Yes

Reviewer #3: Yes

3. Has the statistical analysis been performed appropriately and rigorously? 

Reviewer #2: Yes

Reviewer #3: Yes

4. Have the authors made all data underlying the findings in their manuscript fully available?

Reviewer #2: Yes

Reviewer #3: Yes

5. Is the manuscript presented in an intelligible fashion and written in standard English?

Reviewer #2: (No Response)

Reviewer #3: Yes

6. Review Comments to the Author

Reviewer #2: (No Response)

Reviewer #3: Author should elaborate the data collection sources and the representation of data should be more clear and concise. The authors are advised to add few latest citations, kindly read and cite: a) Smart Contract-Enabled Secure Sharing of Health Data for a Mobile Cloud-Based E-Health System. Appl. Sci. 2023, 13, 3970. https://doi.org/10.3390/app13063970, b) A deep learning approach for facial emotions recognition using principal component analysis and neural network techniques. The Photogrammetric Record, 37, 435–452.c) A secure framework for IoT-based smart climate agriculture system: Toward blockchain and edge computing" Journal of Intelligent Systems, vol. 31, no. 1, 2022, pp. 221-236. https://doi.org/10.1515/jisys-2022-0012

7. PLOS authors have the option to publish the peer review history of their article (what does this mean?). If published, this will include your full peer review and any attached files.

Reviewer #2: No

Reviewer #3: **Yes: **Dr Mahtab Alam

---

## [Author Response · Author response to Decision Letter 1]

21 Mar 2024

Dear Editor and Reviewers:

On behalf of my co-authors, we thank you very much for giving us an opportunity to revise our manuscript, we appreciate editor and reviewers very much for their positive and constructive comments and suggestions on our manuscript entitled “An Evolutionary Game Analysis of Algorithmic Indirect Copyright Infringement from the Perspective of Collusion between UGC Platforms and direct Infringers”. 

We have studied reviewer’s comments carefully and have made revision which marked in the paper. We have tried our best to revise our manuscript according to the comments.Pease find the revised version, which we would like to submit for your kind consideration.Revised portion are marked in yellow in the paper. The main corrections in the paper and the responds to the reviewer’s comments are as flowing:

Responds to Additional Editor Comments:

(1):The introduction provides a good, generalized background of the topic that quickly gives the reader an appreciation of the wide range of applications for this technology. This section helpfully explains the motivation for the research to current and potential funding agencies. However, to make the motivation clearer and to differentiate the paper some more from other applied papers, the author may wish to provide another sentence giving examples of some of the applications of this technology, along with appropriate references.

Response:We are grateful for the suggestion. To be more clear and in accordance with the reviewer concerns, we have added a brief description as follows: "References to scholars using evolutionary game methods to solve tort problems have been added(line:58-60);And papers on the many areas of evolutionary game use(line:122-125).”

(2):There are several instances where assertions are made that are not substantiated with references. These have been noted in the appropriate sections of this paper. it would be better if you provide references to these assertions as this will make your work more concrete.

Author may wish to elaborate the problem formulation section and add some more details which will make the paper look more vital. I do not think any additional graphics are necessary. 

Response:We have listened carefully to your comments. About the assertion ,we have added References.(line:138-141,156-157).Regarding the assertion of replicating the dynamic part of the equation, we cite the proof of Friedman D's study(line:199-203).Regarding the problem formulation section, we have added detailed descriptions. The behavior of the participating subjects is portrayed in more detail.(line:132-136,150-152).

(3):The author may also wish to give a more detailed discussion on blockchain and machine learning implementation.

The author may wish to mention why it is important to leverage blockchain to explain the motivation for his choice of specimens and accompany this with some references to other studies that demonstrate this importance.

Response:We are very grateful for your reminder that brought to my attention the lack of a blockchain as well as a machine learning literature section.We have included references to blockchain and machine learning in the literature review section.(line:79-85)

Responds to Review Comments to the Author:

Response:We have listened very carefully to your suggestions and have enriched the research in this paper by citing the references that you have recommended.These recommended references are located at the 15th, 20th, and 17th position of the citation.

Again, we would like to express our great appreciation to you and reviewers for comments on our paper. Looking forward to hearing from you.

Thank you and best regards.

---

## [Decision Letter · Decision Letter 2]

3 Apr 2024

An Evolutionary Game Analysis of Algorithmic Indirect Copyright Infringement from the Perspective of Collusion between UGC Platforms and direct Infringers

PONE-D-23-30742R2

Dear Yuxuan yu Shen,

We’re pleased to inform you that your manuscript has been judged scientifically suitable for publication and will be formally accepted for publication once it meets all outstanding technical requirements.

Kind regards,

Mudassir Khan, Ph.D

Academic Editor

PLOS ONE

Additional Editor Comments (optional):

Thanks to the authors for the detailed response and additions. I read through the comments and skimmed the revised PDF, and the updates significantly improved the paper. I would be happy to recommend this paper for publication.

Reviewers' comments:

Reviewer's Responses to Questions

**Comments to the Author**

1. If the authors have adequately addressed your comments raised in a previous round of review and you feel that this manuscript is now acceptable for publication, you may indicate that here to bypass the “Comments to the Author” section, enter your conflict of interest statement in the “Confidential to Editor” section, and submit your "Accept" recommendation.

Reviewer #2: All comments have been addressed

2. Is the manuscript technically sound, and do the data support the conclusions?

Reviewer #2: Yes

3. Has the statistical analysis been performed appropriately and rigorously? 

Reviewer #2: Yes

4. Have the authors made all data underlying the findings in their manuscript fully available?

Reviewer #2: Yes

5. Is the manuscript presented in an intelligible fashion and written in standard English?

Reviewer #2: Yes

6. Review Comments to the Author

Reviewer #2: (No Response)

7. PLOS authors have the option to publish the peer review history of their article (what does this mean?). If published, this will include your full peer review and any attached files.

Reviewer #2: No
